# Medial Tibial Stress Syndrome in Novice and Recreational Runners: A Systematic Review

**DOI:** 10.3390/ijerph17207457

**Published:** 2020-10-13

**Authors:** Claudia Menéndez, Lucía Batalla, Alba Prieto, Miguel Ángel Rodríguez, Irene Crespo, Hugo Olmedillas

**Affiliations:** 1Department of Functional Biology, Universidad de Oviedo, 33003 Oviedo, Spain; claumenendez95@gmail.com (C.M.); lbatallanistal@gmail.com (L.B.); aprietin@gmail.com (A.P.); miguerguez95@gmail.com (M.Á.R.); icreg@unileon.es (I.C.); 2Institute of Biomedicine, Universidad de León, 24071 León, Spain; 3Health Research Institute of the Principality of Asturias (ISPA), 33011 Oviedo, Spain

**Keywords:** running, running-related injuries, overuse injury, shin splints, navicular drop

## Abstract

This systematic review evaluates the existing literature about medial tibial stress syndrome (MTSS) in novice and recreational runners. PubMed/MEDLINE, EMBASE, Web of Science, Scopus, SPORTDiscus and CINAHL databases were searched until July 2020. Studies covering risk factors, diagnostic procedures, treatment methods and time to recovery of MTSS in novice and recreational runners were selected. Eleven studies met the inclusion criteria and were included. The risk factors of MTSS are mainly intrinsic and include higher pelvic tilt in the frontal plane, peak internal rotation of the hip, navicular drop and foot pronation, among others. Computed tomography (CT) and pressure algometry may be valid instruments to corroborate the presence of this injury and confirm the diagnosis. Regarding treatment procedures, arch-support foot orthoses are able to increase contact time, normalize foot pressure distribution and similarly to shockwave therapy, reduce pain. However, it is important to take into account the biases and poor methodological quality of the included studies, more research is needed to confirm these results.

## 1. Introduction

Running is one of the most widely practiced physical activities around the world [1]. The number of people enrolling in recreational competitions is growing [2] and the distances they run are becoming longer [3]. However, although the health benefits of running for physically inactive subjects have been demonstrated [4], novice and recreational runners are often affected by running-related injuries [5], which are associated in many cases with training errors characteristic of this population [6,7,8]. In this regard, the lower leg has been proven to be one of the most frequently damaged areas [9], and medial tibial stress syndrome (MTSS) is the main injury occurring in runners [10] at a rate ranging between 13.2% and 17.3% [11].

MTSS consists of a lower leg running injury caused by overuse [12]. It is characterized by exercise-induced pain along the posteromedial border of the distal two-thirds of the tibia and is associated with activity [13]. Despite the fact that its exact cause is unknown, inflammation and muscular traction on the periosteum and painful stress reactions in the bone have been proposed as possible triggering factors [14]. Although most of the published studies have been carried out on professional athletes [15,16], the risk of developing MTSS is noticeably greater among recreational runners [17], where recreational runners are defined as someone with over three months of running experience [5]. Nevertheless, even though adequate management of MTSS is crucial in runners, studies covering this issue are scarce, and it would appear that no reviews have focused on the analysis of MTSS in this population. There is evidence that a higher injury rate in recreational runners and the mechanism of injury of MTSS might be different between recreational and experienced runners [18,19]. This systematic review is intended to assess and summarize the current literature about MTSS in novice and recreational runners.

## 2. Materials and Methods

This systematic review followed the Preferred Reporting Items for Systematic Reviews and Meta-Analyses (PRISMA) statement [20].

### 2.1. Search Strategy and Study Selection

The final literature search was conducted on 1 July, 2020 by two reviewers through five databases: PubMed/MEDLINE, EMBASE, Web of Science, Scopus, SPORTDiscus and CINAHL. No limitations were placed on publication dates, but searches were restricted to English language articles. Details of the search strategy for PubMed/MEDLINE are shown in Appendix A.

The articles that were obtained in the literature search were exported to the Mendeley library, where the duplicate removal tool was used. Two reviewers then independently screened the titles, abstracts and full texts of the articles identified in order to evaluate their possible inclusion in the systematic review. Any disagreements between reviewers were settled through discussion with a third reviewer. Additional studies were identified by supplementing the electronic database search with manually screened references of the included articles and citation tracking of included studies in Scopus.

The studies were required to relate to novice or recreational runners over 16 years old. Recreational runners are defined as someone with over 3 months of running experience, and novice runners are defined as someone with less than 3 months of running experience [5]. Items were excluded if they (a) involved professional runners and athletes from other disciplines, (b) related to military personnel, who are a population with special characteristics, or (c) were of any of the following types: conferences, single-case studies, narrative reviews and systematic reviews.

### 2.2. Data Extraction and Analysis

Various items of data were extracted for each study. These were: authors and year, study design, participants (together with various characteristics such as their sex, age and pathological conditions, if they suffered from one), methodological protocol, main outcome measures and statistically significant results.

Studies were grouped in accordance with their purposes. These were: (1) to study the etiological factors, (2) to assess the diagnostic procedures and (3) to evaluate time to recovery and compare various methods of treatment. Each of these three purposes was assigned an individual section in which the main results were laid out and analyzed descriptively. In view of the heterogeneity of the studies with regard to their targeted outcomes, it did not prove possible to perform a meta-analysis.

### 2.3. Quality Assessment

The quality of the studies was assessed by two researchers using the PEDro scale [21] for randomized controlled trials and the Newcastle–Ottawa scale [22] for cohort, case–control and cross-sectional studies.

## 3. Results

The search strategy retrieved 1379 records. The removal of duplicates ruled 598 studies out of the review process. Thereafter, a further 745 were excluded after consideration of their titles, abstracts or both. Full-text versions of the thirty-six studies remaining were obtained and subjected to further evaluation. After a reading of the full-text copies, twenty-five studies were excluded from this review for the following reasons: (a) fourteen studies were not based on runners or did not specify the type of athletes included, (b) ten studies covered professional/competitive runners and (c) one study included several injuries undifferentiated. At the end of the process, the eleven publications meeting the eligibility criteria were included for qualitative analysis [23,24,25,26,27,28,29,30,31,32,33]. Figure 1 shows the literature search flowchart. The quality scores for the included studies are shown in Table 1 and Table 2. The characteristics of the articles included are displayed in Table 3, Table 4 and Table 5.

### 3.1. Etiological Factors

The overall results are shown in Table 3.

Three studies [25,27,32] provided an analysis of the intrinsic etiological factors responsible for the appearance of MTSS, while one study [30] assessed running distances, an extrinsic etiological factor involved in this pathology. Table 4 summarizes the factors that showed significant differences in at least one study between runners suffering MTSS and controls.

Running kinematics, in both stance and swing phases, are strongly related to the development of MTSS and all the joints in the lower limbs can contribute to the emergence of this pathology. Overpronation of the foot seems to be related to its development, since it has been associated with several biomechanical findings present in MTSS runners (Figure 2).

### 3.2. Diagnostic Procedures

Two studies [24,33] assessed diagnostic procedures for MTSS, involving measurement of the pressure pain threshold with an algometer and high-resolution computed tomography (CT). Results for the diagnosis of MTSS are displayed in Table 5. Although medical history and physical examination, principally through palpation, are crucial both for the evaluation of lower extremity pain and for the diagnosis of MTSS [34], complementary imaging techniques are keys to confirming the presence of this injury.

### 3.3. Treatment Methods and Time to Recovery

Two pieces of research [29,31] studied the time to recovery for MTSS, while three [23,26,28] examined treatment methods for MTSS. Table 6 summarizes the evidence gathered on the treatment of MTSS in recreational runners.

## 4. Discussion

MTTS is the most frequent injury diagnosed in runners. Although novice and recreational runners are often affected, this is the first systematic review focus on this topic. We have summarized the intrinsic (pelvic drop, peak hip internal rotation knee flexion, female sex, navicular drop, abductory twist during gait, apropulsive gait, pronation in a static position and early heel lift during the stance phase of gait) and extrinsic factors (walking distance), calling for specific preventive strategies. We have also collected the studies covering diagnosis (pressure algometry and CT) and treatment (foot orthoses and shockwave therapy), providing valuable information for clinicians and physicians. However, the majority of the included studies have a lower level of evidence, so this information should be interpreted carefully.

Higher values for navicular drop, an indicator of pronation of the foot [35], have been observed in runners with MTSS relative to their uninjured peers [32,36]. Abductory twist and early heel lift have also been described as compensatory mechanisms leading to overpronation and an apropulsive gait in running [27], in which the load on the posterior tibial tendon is increased [37] and favors overuse injuries [11]. Furthermore, both a higher pelvic drop and peak internal rotation of the hip and lower knee flexion predispose runners to suffer MTSS during the stance phase in running [25] and have been linked with other overuse injuries in this group [38].

An increased pelvic drop has been associated with a medial displacement of the center of mass of the body [39], which leads to knee valgus and subtalar joint pronation [40], while high values for internal rotation of the hip have also been linked to overpronation of the foot [41]. Furthermore, the stress caused by excessive pronation during motion in the muscles responsible for sustaining the arch of the foot may increase the maximum voluntary isometric contraction torque of the first metatarsophalangeal joint in plantar flexion, which has been observed in runners with a history of MTSS [42]. In addition to all the factors analyzed, it is worth mentioning that none of the studies included evaluated body mass index, despite its strong association with the occurrence of injuries in beginner runners [43].

Moreover, Nielsen et al. [30] showed that running distance is an extrinsic factor associated with MTSS. They divided runners into three groups based on their increase or decrease in weekly running distances and observed that those expanding their runs more (>30%) had a higher prevalence rate for this injury. These findings concur with the fact that a long training distance each week and a history of previous injuries are predisposing factors for injuries [9]. This may be the consequence of the decreased running speed experienced by runners when they confront longer distances and become tired, increasing the cumulative number of steps [43] and overloading the knee joint through this reduction in speed [44].

Finally, with regard to sex, Loudon and Reiman [25] observed a higher pelvic drop in women with MTSS in comparison with men while running. This agrees with others who have reported that females are at more risk of MTSS than males [45,46]. Hence, differences in running kinematics between the two sexes have been suggested as factors responsible for this finding, although the reason that females are at increased risk of this pathology remains unknown.

In view of the close relationship between the biomechanics of running and the incidence of injuries [47,48], specific training in running technique should be carried out to reduce the risk of injury. Indeed, army recruits at risk of MTSS who were submitted to a 26-week gait retraining protocol consisting of plantar pressure system biofeedback. Verbal orders and corrections had a lower incidence rate of MTSS at the end of the protocol. It is also worth noting that this procedure was complemented with strength, flexibility and neuromuscular control exercises, evidencing the importance of designing multicomponent protocols for this purpose [49].

Aweid et al. [24] found that a pressure algometer was a well-tolerated and reliable method for assessing the pain pressure threshold in nine runners with MTSS and twenty showing no symptoms. These results agree with those of others who have reported acceptable reliability for this procedure [50,51], and may support the use of pressure algometry for monitoring the rehabilitation process.

Gaeta et al. [33] carried out CT examinations on eleven runners with MTSS, twenty asymptomatic runners and control peers not involved in a sport, classified according to their loss of bone mass. Good diagnostic accuracy was demonstrated for high-resolution CT and, although more studies are required, CT may be useful to detect stress-induced bone remodeling. This would go to support the hypothesis that MTSS consists of a bone stress reaction, rather than the theory associating this injury with an inflammation of the periosteum arising from excessive traction [11]. In relation to this, Magnusson et al. [52] observed lower bone mineral density in eighteen adult male athletes with long-standing MTSS, as compared with asymptomatic controls, although no statistically significant differences in tibial bone density were recorded between eleven athletes with MTSS and a healthy control group [53]. This may have been an outcome of a shorter history of injury.

The main goal of MTSS treatment is to relieve pain and allow sufferers to return to practicing sports without discomfort [54]. However, there is currently a lack of evidence to allow decisions about which techniques are the most appropriate for rehabilitation and readaptation [55]. Loudon and Dolphino [28] designed a three-week protocol that combined the use of an off-the-shelf basic foot orthotic (BFO) during walking hours with a standard gastrocnemius and soleus stretching program twice a day for three weeks. They observed reductions both in levels of pain (50%) and in the duration of symptoms in 65% of participants, in addition to an enhanced quality of life. However, from this study, we cannot conclude that the isolated use of BFO is responsible for the improvements. On these lines, runners with MTSS who use bilateral arch-supporting foot orthoses have shown improvements in foot pressure, decreasing the total contact time and force over time in the midfoot area, which was translated to the fifth metatarsal region. Hence, these subjects noted lateral pressure displacement at the forefoot flat, heel-off and push-off phases and medial displacement during the forefoot contact phase during running, so this method may be an effective alternative to treat MTSS in recreational runners [26]. Nevertheless, the normalized foot pressure distribution patterns during running through arch-support orthoses do not involve a reduction in the level of pain in these subjects. Although the mechanisms through which the use of foot orthoses can bring benefits remain unknown [56], these results suggest that a correction in overpronation might be key, since it is associated with MTSS symptoms [57]. Thus, as concluded by Newman et al. [45] and supported by other studies [32,36], a strong navicular drop seems to be the factor responsible for MTSS. This can be corrected by orthotic treatments [58], which have proved to reduce foot eversion [59,60].

Focused shockwave therapy (SWT) on MTSS has not shown any significant differences when a standard dose of 1450 millijoules per square millimeter (mJ/mm^2^) and a sham dose (70 mJ/mm^2^) were compared [23]. Patients with MTSS have seen faster recovery and decreased levels of pain following SWT compared to exercise protocols [61,62,63], although the authors reporting this did not evaluate different doses, making it impossible to determine whether the dosage plays any relevant part in the success of this intervention. 

With regard to recovery times, similar average values have been reported in runners both novice and recreational (70 and 72 days, respectively) [29,31,49]. However, in the only randomized control trial performed with different treatment protocols in recreational runners, runners took an average of six months to recover sufficiently to complete an 18-minute run [64].

### 4.1. Clinical Relevance

Novice and recreational runners do not have the same resources as professional runners to enhance performance at their disposal. This situation highlights the need for specific strategies to be implemented in this population to reduce the incidence of injuries derived from impaired biomechanics. We consider that there are three main key points that would be essential for novice and recreational runners (the first two points for prevention and the third for rehabilitation):To undergo a biomechanical analysis of running movements to identify risk factors for injury.To perform a specific running technique to improve running kinematics. This training protocol should be accompanied by strength and neuromuscular control exercises.To establish a gradual running program with proper recovery times, in order to manage pain and prevent injury recurrence.

### 4.2. Quality of the Studies and Limitations

The studies included in this systematic review present some design biases that prevent us from establishing solid conclusions. It is worth mentioning that only the study by Newman et al. [23] is a randomized controlled trial (level of evidence II), while the rest of the studies have a lower level of evidence. Therefore, when interpreting the results, we have to be aware of this circumstance, which is a major limitation in order to obtain solid evidence from this systematic review. We encourage further high-quality research on this specific population to provide clear conclusions.

## 5. Conclusions

The main factors for the development of MTSS in novice and recreational runners seem to be intrinsic and are commonly related to a biomechanical origin. To achieve a diagnosis, CT scan has shown to be an accurate method, detecting stress-induced bone remodeling. Pressure algometry is a complementary tool for assessing pain, the injury state and the rehabilitation process. The approximate recovery time is sixteen to eighteen weeks. Treatment methods include arch-support orthoses, which induce positive effects on the foot pressure distribution, and SWT, which might be able to reduce the level of pain. We have to bear in mind the poor methodological quality of the included studies, which prevents us from obtaining solid evidence about MTSS in novice and recreational runners.

## Figures and Tables

**Figure 1 ijerph-17-07457-f001:**
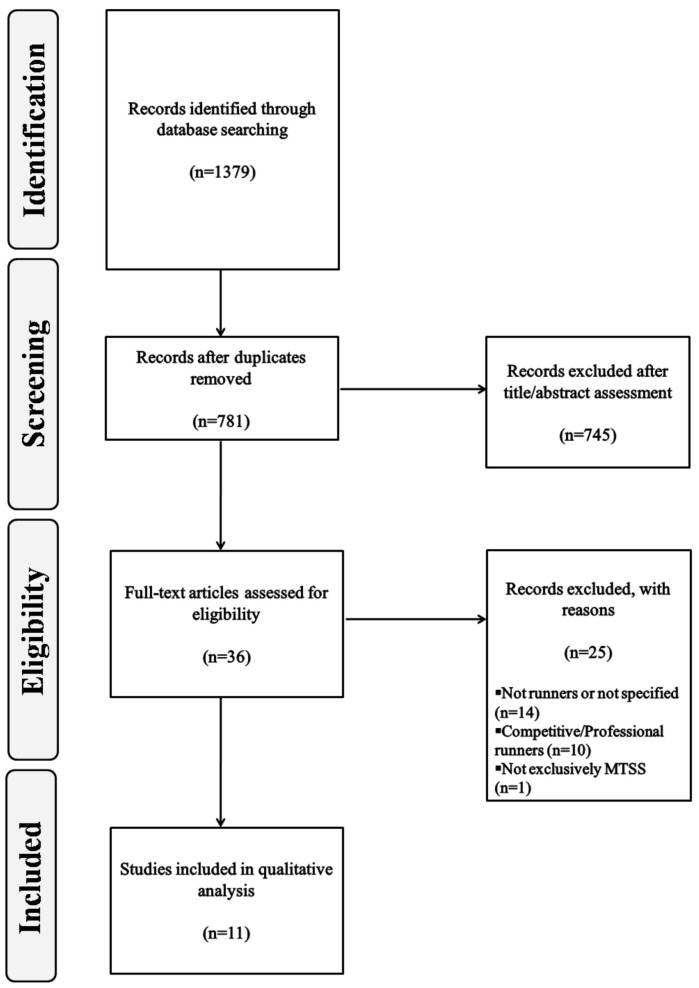
Flow diagram of study selection process. MTSS: medial tibial stress syndrome.

**Figure 2 ijerph-17-07457-f002:**
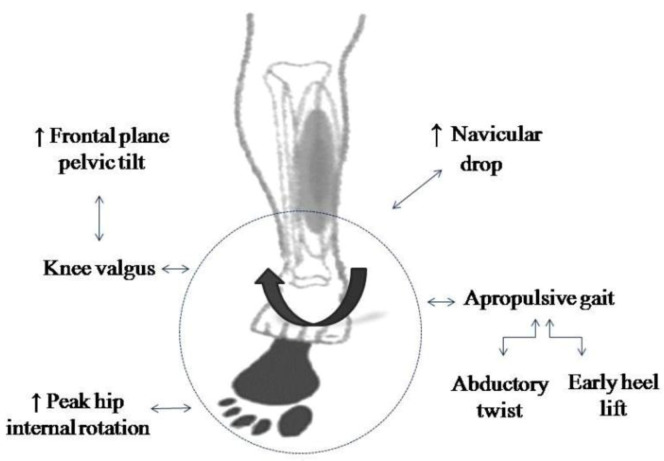
Factors commonly associated with foot overpronation.

**Table 1 ijerph-17-07457-t001:** PEDro score for randomized controlled trials and quasi-experimental studies.

Study	PEDro Score Distribution											
	1	2	3	4	5	6	7	8	9	10	11	Total PEDro Score
Newman et al., 2017 [23]	1	1	1	1	1	─	1	1	1	1	1	9

1—Eligibility criteria; 2—Random allocation; 3—Concealed allocation; 4—Baseline comparability; 5—Blind subjects; 6—Blind therapists; 7—Blind assessors; 8—Adequate follow-up; 9—Intention-to-treat analysis; 10—Between-group comparisons; 11—Point estimates and variability. A “1” indicates a “yes” score, and a dash indicates a “no” score.

**Table 2 ijerph-17-07457-t002:** Newcastle–Ottawascale scores for observational studies.

Study Design	Study	Newcastle–Ottawa Scale								
		Selection				Comparability	Exposure /Outcome			Total Score
		1	2	3	4	1	1	2	3	
1	Aweid et al., 2014 [24]	★			★	★★	★	★		★★★★★(6)
	Loudon and Reiman, 2012 [25]	★			★	★★	★	★	★	★★★★★★★(7)
	Naderi et al., 2019 [26]	★	★	★	★	★★	★	★	★	★★★★★★★★★(9)
	Tweed et al., 2008 [27]	★	★	★	★	★★	★	★		★★★★★★★★(8)
2	Loudon and Dolphino, 2010 [28]	★		★	★	★★	★	★		★★★★★★★(7)
	Mulvad et al., 2018 [29]			★		★		★		★★★(3)
	Nielsen et al., 2014 [30]	★	★	★		★★	★	★		★★★★★★★(7)
	Nielsen et al., 2014 [31]	★	★	★		★	★	★		★★★★★★(6)
	Raissi et al., 2009 [32]				★		★	★		★★★ (3)
3	Gaeta et al., 2006 [33]			★	★★	★	★★	★	N/A	★★★★★★★(7)

Case–control studies: Selection (maximum ★★★★): (1) Is the case definition adequate?, (2) Representativeness of the cases, (3) Selection of controls, (4) Definition of controls; Comparability (maximum ★★): (1) Comparability of cases and controls on the basis of the design or analysis; Exposure (maximum ★★★): (1) Ascertainment of exposure, (2) Cases and controls: same ascertainment method, (3) Cases and controls: same non response date. Cohort studies: Selection (maximum ★★★★): (1) Representativeness of the exposed cohort, (2) Selection of the non exposed cohort, (3) Ascertainment of exposure, (4) Demonstration that outcome of interest was not present at the start of the study; Comparability (maximum ★★): (1) Comparability of cohorts on the basis of the design or analysis; Outcome (maximum ★★★): (1) Assessment of outcome, (2) Was follow-up long enough for outcomes to occur, (3) Adequacy of the follow-up of cohorts. Cross-sectional studies: A modified Newcastle–Ottawa scale for cross-sectional studies was used. Selection (maximum ★★★★★): (1) Representativeness of the sample, (2) Sample size, (3) Non respondents, (4) Ascertainment of the exposure (risk factor); Comparability (maximum ★★): (1) The subjects in different outcome groups are comparable, based on the study design or analysis. Confounding factors are controlled; Outcome (maximum ★★★): (1) Assessment of outcome; (2) Statistical test. **N/A:** not applicable.

**Table 3 ijerph-17-07457-t003:** Summary of studies analyzing etiological factors.

Study	Design Level of Evidence (NHMRC)	Participants	Methodology	Results (*p* <0.05)
Loudon and Reiman, 2012 [25]	Case–control III-3	28 runners ▪ 14 runners with a history of unilateral MTSS (8♂/6♀) ∗ Age: 29.2 ± 5.96 ∗ Training miles: 21.07 ± 16.42 ▪ 14 control runners (8♂/6♀) ∗ Age: 26.5 ± 5.39 ∗ Training miles: 17.29 ± 13.96	▪ Kinematic analysis of the pelvis, hip and knee during treadmill running	▪ *↑* PD and peak hip IR, and ↓knee flexion ▪ *↑* PD in the injured limbin♀ vs. ♂ in the MTSS group
Nielsen et al., 2014 [30]	Cohort, prospective II	873 runners ▪ 202 injured (106♂/96♀) ∗ Age: 39.0 ± 10.3 ▪ 671 injury-free runners (335♂/336♀) ∗ Age: 36.7 ± 10.2	▪ Runners were divided into 3 groups according to progression in weekly running distance (wrd)	▪ >30% wrd *↑* prevalence in MTSS
Raissi et al., 2009 [32]	Cohort, prospective II	66 runners (21♂/45♀) ∗ Age: 63.6 ± 7.1 ∗ Height (m): 1.68 ± 0.9	Goniometric measurement: intercondylar and intermalleolar interval, Q angles, tibiofemoral angle, rear foot alignment (Achilles or calcaneal angle), tibial alignmentandND test. ▪ Tape measure measurement: leg length. 17-week running program	▪ 16 runners developed MTSS bilaterally ▪ *↑* ND in MTSS runners
Tweed et al., 2008 [27]	Case–control III-3	40 runners ∗ Age: 18–56 years ▪ 28 runners with MTSS (16♂, 12♀) ▪ 12 control runners (7♂, 5♀)	▪ Foot type (Foot Posture Index) Talocrural joint dorsiflexion and the range of motion of the 1st MTPJ (tractography) ▪ Static and dynamic gait kinematic (treadmill running)	▪ Predictors of MTSS: Difference between neutral and relaxed calcaneal; ROM of the talocrural joint;early heel lift;abductorytwistandapropulsive gait

IR: Internal Rotation; MTSS: Medial Tibial Stress Syndrome; ND: Navicular Drop; NHMRC: National Health and Medical Research Council; PD: Pelvic Drop; vs.: versus.↑: higher; ↓: lower. ♂: man; ♀: woman.

**Table 4 ijerph-17-07457-t004:** Main statistically significant factors of Medial Tibial Stress Syndrome.

Factors	Type		Study
	Intrinsic	Extrinsic	
↑ Pelvic drop	✓		Loudon and Reiman, 2012 [25]
↑ Peak hip internal rotation	✓		
↓ Knee flexion	✓		
Female sex	✓		
↑ Walking distance		✓	Nielsen et al., 2014 [30]
↑ Navicular drop	✓		Raissi et al., 2009 [32]
Early heel lift during the stance phase of gait	✓		Tweed et al., 2008 [27]
Abductory twist during gait	✓		
Apropulsive gait	✓		
Pronation in a static position	✓		

↑: higher; ↓: lower.

**Table 5 ijerph-17-07457-t005:** Summary of studies analyzing diagnostic procedures.

Study	Design Level of Evidence (NHMRC)	Participants	Methodology	Results (*p* < 0.05)
Aweid et al., 2014 [24]	Case–control III-3	29 runners ▪ 9 runners with MTSS (6♂/3♀) ∗ Age: 28.2 ± 9.2 ▪20 asymptomatic runners (10♂/10♀) ∗ Age: 24 ± 3.0	▪ Pressure algometry - Measurements along the medial border of the tibia	Runners with MTSS: ▪ Males had >PPT than females at the 6/9 area of the tibia Comparison between groups: ▪ *↑* PPT for MTSS runners at the 3/9 site
Gaeta et al., 2006 [33]	Cross-sectional IV	41 participants ▪ 20 asymptomatic runners (14♂/6♀) ∗ Age: 18–25 ▪ 10 nonathletes volunteers (7♂/3♀) ∗ Age: 20–26 ▪ 11 runners with MTSS (7♂/4♀) ∗ Age: 20–24	▪ CT Tibias were classified according to the tibial cortex: - Type 0: no abnormality - Type 1: reduced cortical attenuation - Type 2: cortical osteopenia (MTSS)	▪ Asymptomatic runners - Type 0: 22 - Type 1: 13 - Type 2: 5 ▪ Runners with MTSS -Type 0: 2 - Type 1: 3 - Type 2: 3 (painless) and 14 (pain) ▪ Symptomatic tibias with CT abnormalities: 100% ▪ Asymptomatic tibias with CT abnormalities: 16.6%

CT: Computed Tomography; MTSS: Medial Tibial Stress Syndrome; NHMRC: National Health and Medical Research Council; PPT: Pain Pressure Threshold. ♂: man; ♀: woman.

**Table 6 ijerph-17-07457-t006:** Summary of studies analyzing treatment methods and time to recovery.

Study	Design Level of Evidence (NHMRC)	Participants	Methodology	Results (*p* <0.05)
Loudon and Dolphino, 2010 [28]	Case series IV	23 runners with MTSS (12♂/11♀) ∗ Age: 28.1 ± 5.9	Initial measurements: ▪ Alignment: ND test and talocrural dorsiflexion ROM ▪ Pain (NPRS) Treatment (3 weeks): ▪ Off-the-shelf BFO ▪ Home stretching program Follow-up: ▪ GRC questionnaire by email (Post 7 and 21 days)	▪ ↓ Duration of the symptoms: 44% ♀and 83% ♂ ▪ ↓ Pain level in the successful group ▪ Improvement in GRC in the successful group (post 21 days)
Mulvad et al., 2018 [29]	Cohort III-2	112 injured runners (30♂/82♀) ∗ Age: 41.4(21–63) ▪18 runners with MTSS (16%)	▪ Clinical examinations	▪ MTSS had the longest time to recovery (70 days)
Naderi et al., 2019 [26]	Case–control III-2	100 runners (100♂) ▪ 50 runners with MTSS ∗ Age: 21.9 ± 2.4 ▪ 50 healthy runners ∗ Age: 21.1 ± 2.5	▪ Arch-support full-length footorthoses in the shoes ▪ Pain measurement (algometry, VAS) ▪ Foot postures (FPI-6) ▪ Dynamic pressure distribution during running (stance phase)	After using arch-support foot- orthosesin MTSS runners: ▪ ↓ Total contact time ▪ ↓ Absolute impulse in the midfoot region and ↑peak pressure and absolute impulse underneath the M5 region ▪ Pressure distribution was laterally at FFF and HO ▪ Pressure displacement was shifted from lateral to medial during FFCP and from medial to lateral during FFPOP ▪ The X-component of the COP was more medially at FFF and was displaced to lateral side during FFPOP ▪ The X-component of the COP was laterally at FFF and the lateral COP displacement ↓during FFPOP
Newman et al., 2017 [23]	Randomized controlled trial II	24 runners with MTSS ▪ Experimental group: 12 (5♂/7♀) ∗ Age: 34 ± 11 ▪ Control group: 12 (5♂/7♀) ∗ Age: 36 ± 9	Treatment: ▪ Experimental group: standard dose shockwave therapy (total = 1450 mJ/mm2) ▪ Control group: sham dose shockwave therapy (total = 70 mJ/mm2)Measurements: ▪ Pain: pressure algometry and during running (NRS) ▪ Self GROC questionnaire Follow-up: 10 weeks	▪ No significant differences between groups
Nielsen et al., 2014 [31]	Cohort III-2	254 injured runners ▪ 38 runners with MTSS (15%), (18♂/20♀)	▪ Clinical examinations ▪ Prospective follow-up to determine time to recovery	▪ Mean time to recovery of 72 days

BFO: Basic Foot Orthotic; COP: Center of Pressure; FFCP: Forefoot Contact Phase; FFF: Forefoot Flat; FFPOP: Forefoot Push-Off Phase; FPI-6: Foot Posture Index; GRC: Global Rating of Change; GROC: Global Rating of Change; mJ: millijoules; NHMRC: National Health and Medical Research Council; NPRS: Numerical Pain Rating Scale; MTSS: Medial Tibial Stress Syndrome; M5: 5th metatarsal; ND: navicular drop; NRS: Numerical Rating Scale; ROM: Range of Motion; VAS: Visual Analog Scale. ♂: man; ♀: woman.

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
