# Peer review of "Medial Tibial Stress Syndrome in Novice and Recreational Runners: A Systematic Review"

_ijerph, 2020, doi:10.3390/ijerph17207457_

Round 1
Reviewer 1 Report
Thanks for allowing me to review this re-submission. I think many of the issues have been suitably addressed with just a few minor changes needed:
Regarding the comment of impulse – I recognise the definition of ‘impulse’ in mechanics, but still feel that there are multiple other definitions for impulse (such as an electrical impulse), it may be better to change to ‘momentum’, or the more descriptive ‘force over time’. But I think ultimately that will depend on the authors and editor’s preference.
Line 236 – change ‘lesser’ to ‘decreased’ to improve grammar
Line 245-246 – This sentence doesn’t make sense – “Novice and recreational runners do not dispose of the means that their professionals peers do to enhance performance”
Do you mean they don’t have access to the same resources?
Maybe something like “have at their disposal the same resources as…..”
I also think that the clinically relevant findings need some changes regarding grammar and comprehension. i.e.-
- To undergo a biomechanical analysis of running movements to identify risk factors for injury. Nonetheless, the lack of knowledge of the determined causes of MTSS makes it difficult to design optimal prevention protocols.
- To undertake specific training in running technique training to address lower limb biomechanics where required. As we have mentioned above, this protocol should be accompanied by strength and neuromuscular control exercises.
- To establish gradually scaled appropriate return to sport rehabilitation programs with sufficent recovery times and load management following MTSS, in order to manage pain and prevent injury recurrence.
Line 260-261 - It is worth mentioning that only the study by Newman et al, 2017 [31] is a randomized controlled trial (level of evidence II), while the rest of the studies have a lower level of evidence.
Author Response
Reviewer 1:
Thanks for allowing me to review this re-submission. I think many of the issues have been suitably addressed with just a few minor changes needed:
Regarding the comment of impulse – I recognise the definition of ‘impulse’ in mechanics, but still feel that there are multiple other definitions for impulse (such as an electrical impulse), it may be better to change to ‘momentum’, or the more descriptive ‘force over time’. But I think ultimately that will depend on the authors and editor’s preference.
Thank you for your suggestion. We have modified this term in the text, but we have keep it on the tables (Statistically significant results (p<0.05), since the original authors used it.
Line 236 – change ‘lesser’ to ‘decreased’ to improve grammar
Thank you, for the correction. We have change it.
Line 245-246 – This sentence doesn’t make sense – “Novice and recreational runners do not dispose of the means that their professionals peers do to enhance performance”
Do you mean they don’t have access to the same resources?
Maybe something like “have at their disposal the same resources as…..”
We agree. We have changed the sentence as the reviewer suggest.
I also think that the clinically relevant findings need some changes regarding grammar and comprehension. i.e.-
Thank you. We have rewritten the text:
- To undergo a biomechanical analysis of running movements to identify risk factors for injury.
- To perform specific running technique to improve running kinematics. This training protocol should be accompanied by strength and neuromuscular control exercises.
- To establish gradual running program with proper recovery times, in order to manage pain and prevent injury recurrence.
Line 260-261 - It is worth mentioning that only the study by Newman et al, 2017 [31] is arandomized controlled trial (level of evidence II), while the rest of the studies have a lower level of evidence.
Done, thank you.
Reviewer 2 Report
I would like to start with an acknowledge to authors for the effort of preparing and submitting this paper. The topic is interesting and relevant for a big group of recreational runners. For the most part I find the content prepared appropriately, however some issues need some polishing and clarification to be better understandable for a reader, and to prepare better version of the manuscript.
Please below find my comments to different parts of the manuscript.
Introduction
It might be necessary to underline more the difference among the current review and other previously published systematic review regarding the topic of MTSS, that are cited in the current study. What has been established up till now, and what is new in current study?
Materials and Methods
Although it is stated that the literature search was conducted on July 2020, it might be relevant to add the days, e.g. 1-31.07 (line 54).
Why authors limited their search strategy only to “medial tibial stress syndrome” whereas the syndrome has some other description, e.g. "shin splints", "shin pain"?
If the authors decided to limit the review only to recreational and novice runners, it is necessary to clear the definition: who is the “novice runner” and who is the “recreational runner”. The inclusion criteria might be very helpful for precise interpretation of the results, since a lot of systematic reviews have been published in terms of MTSS in runners. And the description of exclusion criteria seems to be very broad.
Results and Discussion
In my opinion these two sections should be rewritten. Authors should provide separately section results section and discussion paragraph. In results section also analysis of quality of the papers should be thoroughly elaborated. I am sorry but I also find current tables very difficult to analyse. Maybe it would be more adequate to redefine tables and try to collect the main data in each of table.
Abstract
Although it is at the beginning of the manuscript, I decided to include my comments on abstract et the end of this review. I hope my above mention comments are necessary to underline my doubts of the current version of the abstract. I think, regarding all limitations of the study (including PICO strategy) that the abstract presents overestimated information.
“Abstract: This systematic review evaluates the existing literature about medial tibial stress syndrome (MTSS) in novice and recreational runners.” (lines 13-14).
No definition is provided in terms of novice and recreational runners. What is the difference between this groups? How often the runner can run to be classified only as the recreational? Or… is it only the issue of “being a member of any formal club/federation”? Why authors decided to include runners who are at least 16 years old? Why not over 18, as in many countries the legal age of adults is 18 years?
“The average time to recovery was approximately 16-18 weeks after the start of the pathology.” )lines 23-24). Although 11 studies were included, three of them were with very low quality, so I would not rather provide any further recommendation based also on them.
“Nonetheless, bearing in mind the biases and the poor methodological quality of the included studies, further research is needed to confirm these results.” Lines 24-25. I think this statement should be more underlined, since the quality of included studies are low.
I hope with this feedback I encourage authors for considering my comments to improve the quality of authors' manuscript.
Author Response
I would like to start with an acknowledge to authors for the effort of preparing and submitting this paper. The topic is interesting and relevant for a big group of recreational runners. For the most part I find the content prepared appropriately, however some issues need some polishing and clarification to be better understandable for a reader, and to prepare better version of the manuscript.
Thank you
Please below find my comments to different parts of the manuscript.
Introduction
It might be necessary to underline more the difference among the current review and other previously published systematic review regarding the topic of MTSS, that are cited in the current study. What has been established up till now, and what is new in current study?
Thank you for your analysis. We agree with the reviewer that some other researches have previously studied this topic. However, we pretend to cover the whole spectrum (diagnostic, prevention, risk factors and treatment) of existing studies on medial tibial stress syndrome exclusively in novice and recreational runners, since the increasing number of people develop this injury in the beginning of the running practice. The running population, especially in long distance events, is increasing, with the risk of running-related injuries. Novice runners seem to be the most vulnerable group, with a much higher risk of injury (Videbaek S, et al., 2017). The current literature does not count with a systematic review about MTSS in this population, since as we mention in the introduction, the studies are scarce in this population and the vast majority only include professional runners/athletes.
We update the review up July 2020, since last review was published in 2017. Nevertheless, Reinking et al, 2017, performed the search analysis “Between October 2011 and May 2012”. Thus, the great systematic reviews published on same topic only focus in risk factors, and different population is included in the manuscripts (Reinking et al, 2017, Newman et al, 2013 and Winkelmann et al, 2016).
Therefore, as no reviews have been performed on this exact topic, we think that it is interesting given the large population of novice and recreational runners is growing year by year.
Materials and Methods
Although it is stated that the literature search was conducted on July 2020, it might be relevant to add the days, e.g. 1-31.07 (line 54).
Thank you, we have added the day
Why authors limited their search strategy only to “medial tibial stress syndrome” whereas the syndrome has some other description, e.g. "shin splints", "shin pain"?
- Reviewer is right. When we plan to perform the search we included this terms, but after MESH term “medial tibial stress syndrome”, the database include “Shin Splints, Shin Splint”, since 2011, and is define as a shin pain. So all the potential studies are included in our search strategy.
If the authors decided to limit the review only to recreational and novice runners, it is necessary to clear the definition: who is the “novice runner” and who is the “recreational runner”. The inclusion criteria might be very helpful for precise interpretation of the results, since a lot of systematic reviews have been published in terms of MTSS in runners. And the description of exclusion criteria seems to be very broad.
Thank you. Regarding the definition of recreational runners, a well-designed systematic review and meta-analysis by Videbæk et al (Videbæk S, Bueno AM, Nielsen RO, Rasmussen S. Incidence of Running-Related Injuries Per 1000 h of running in Different Types of Runners: A Systematic Review and Meta-Analysis. Sports Med. 2015;45(7):1017-1026), establishes a cut-off around 8-13 weeks of running experience for distinguishing a novice runner from a recreational runner. We have based on this study and we have established a 3-month running experience as some criteria for be considered a novice or recreational runner. We have included theses definitions in this section. (72-74)
Results and Discussion
In my opinion these two sections should be rewritten. Authors should provide separately section results section and discussion paragraph. In results section also analysis of quality of the papers should be thoroughly elaborated. I am sorry but I also find current tables very difficult to analyse. Maybe it would be more adequate to redefine tables and try to collect the main data in each of table.
We understand the opinion of the reviewer, but since the beginning of the redaction of the manuscript this was performed together. The current journal “IJERPH” allows this:
“Discussion: Authors should discuss the results and how they can be interpreted in perspective of previous studies and of the working hypotheses. The findings and their implications should be discussed in the broadest context possible and limitations of the work highlighted. Future research directions may also be mentioned. This section may be combined with Results.”
Based on this and bearing in mind the structure of our review, which is heterogeneous regarding the thematic (diagnostic procedures, treatment methods, etc), we firmly believe that combining both sections is the optimal way to facilitate reading and understanding.
The information in the tables has been modified and summarized in order to make it more understandable.
Abstract
Although it is at the beginning of the manuscript, I decided to include my comments on abstract et the end of this review. I hope my above mention comments are necessary to underline my doubts of the current version of the abstract. I think, regarding all limitations of the study (including PICO strategy) that the abstract presents overestimated information.
“Abstract: This systematic review evaluates the existing literature about medial tibial stress syndrome (MTSS) in novice and recreational runners.” (lines 13-14).
No definition is provided in terms of novice and recreational runners. What is the difference between this groups? How often the runner can run to be classified only as the recreational? Or… is it only the issue of “being a member of any formal club/federation”? Why authors decided to include runners who are at least 16 years old? Why not over 18, as in many countries the legal age of adults is 18 years?
We have just justified this point in the introduction section on the line 46, and after your suggestion we added in the methods section. We consider that it is possible from the age of 16 to differentiate between runners who follow a structured training program train and recreational runners. Additionally, we have observed in clinical practice that several runners below 18 years old where diagnosed with MTSS in the beginning of the running practice.
“The average time to recovery was approximately 16-18 weeks after the start of the pathology.” )lines 23-24). Although 11 studies were included, three of them were with very low quality, so I would not rather provide any further recommendation based also on them.
Thank you for your suggestion We have eliminated this sentence.
“Nonetheless, bearing in mind the biases and the poor methodological quality of the included studies, further research is needed to confirm these results.” Lines 24-25. I think this statement should be more underlined, since the quality of included studies are low.
However, it is important to take into account the biases and poor methodological quality of the included studies, more research is needed to confirm these results.
Thank you for your suggestion. We have modified the sentence.
I hope with this feedback I encourage authors for considering my comments to improve the quality of authors' manuscript.
Thank you for your help.
Reviewer 3 Report
The authors present a thoroughly designed systematic review which adheres to the rules of the PRISMA statement. The existing literature (mainly low level of evidence) has been consistently processed. Due to the quality of the selected studies, there are partially significant differences in data, e.g. for the RTS/recovery times.
"similar average values have been reported in runners both novice and recreational (70 and 72 days, respectively) [29, 33,65]. However, in the only randomized control trial performed with different treatment protocols in recreational runners, took an average of 6 months to recover"
Basic background knowledge, diagnostic and therapeutic measures are discussed in detail.
Moderate English language changes are still required.
Author Response
The authors present a thoroughly designed systematic review which adheres to the rules of the PRISMA statement. The existing literature (mainly low level of evidence) has been consistently processed. Due to the quality of the selected studies, there are partially significant differences in data, e.g. for the RTS/recovery times.
"similar average values have been reported in runners both novice and recreational (70 and 72 days, respectively) [29, 33,65]. However, in the only randomized control trial performed with different treatment protocols in recreational runners, took an average of 6 months to recover"
Basic background knowledge, diagnostic and therapeutic measures are discussed in detail.
Thank you for your review.
Moderate English language changes are still required.
Round 2
Reviewer 2 Report
Dear Authors,
Thank you for all your comments and for all corrections that you made in the revised version of the manuscript.
I suggest verifying whether all included papers were allocated to proper tables: table 1 or table2. Since, paper by Loudon and Dolphino is not designed as the randomized controlled trial it is not proper to include this paper in table 1. Moreover, authors provide more information about the study design of this paper (vide table 6), so the main allocation must be corrected, and other publication should be verified in terms of the study design.
Thank you for adding the discussion section. However, usually, the first paragraph in the discussion section should provide information on the main aim of this systematic review and main findings. I advise authors to analyse some papers already published in the International Journal of Environmental Research and Public Health e.g. https://www.mdpi.com/1660-4601/17/8/2744 or https://www.mdpi.com/1660-4601/17/11/3925
I am open for any further discussion. Thank you in advance for your cooperation.
Author Response
Dear reviewr, Thank you for your valuable suggestion.
The changes have been highlighted in yellow in the manuscript.

This manuscript is a resubmission of an earlier submission. The following is a list of the peer review reports and author responses from that submission.
Round 1
Reviewer 1 Report
Abstract:
Ln 24: CT for confirming the diagnosis? what's the evidence
Ln26: Whats are the other treatments as activity modification, physical therapy etc?
Introduction
Ln 40: MTSS unlike Stress fracture is not localised--please reframe the sentence
Ln: 46: The statement may be incorrect as the ref 45 used in this paper: is on the risk factors of MTSS in runners- Systematic review
Materials:
Good description flow
Studies included seem not to cover the whole spectrum: eg Ref 45 is not included. Similarly other studies from literature are missed
Inclusion criteria in this study had:
"All participants in all papers were runners or played sports that were running-based."
Results
Ln 155: Etiological factors:
Extrinsic factors not described which form an important component of risk factors in MTSS
3.3 Diagnostic procedures:
no description on History and Physical examination
comment on CT are misleading as the author have not commented on any diagnostic signs of MTSS on CT.
Osteopenia in runners? what's the prevalence in this population
Pressure altimeter: is it diagnostic?
Ln214: confusing line, CT examination on eleven runners, twenty having no symptoms"
Ln 217: to make a conclusion after one study
Ln219- 222: comparison is being made with athletes to derive conclusion in non athletes---inadequate
Treatment: inadequate and Incomplete methods: is this because inappropriate study inclusion?
orthotics: Can we generalise the advice?- unlikely
What about extrinsic factor management
what about systemic reasons
The authors mention about osteopenia-- what about its management
Conclusion:
Misleading especially with diagnosis with CT
The author talks bout Hx an physical management-- but no specific mention in the text
SWT: no study quoted for convincing evidence on this treatment
What
Author Response
Abstract:
Ln 24: CT for confirming the diagnosis? what's the evidence
Thank you very much for your suggestion. As there no exist solid evidence, we have modified the sentence: “Computed Tomography (CT) and pressure algometry may be valid instruments to corroborate the presence of this injury and confirm the diagnosis.”(Line 23).
Ln26: Whats are the other treatments as activity modification, physical therapy etc?
Since we have included only studies referring to novice and recreational runners and other treatments do not appear in these studies, we have not included in the review.
Introduction
Ln 40: MTSS unlike Stress fracture is not localised--please reframe the sentence
Thank you for your clarification. We have changed the sentence: “MTSS consists of a lower leg running injury caused by over-use [12]. It is characterized by exercise-induced pain along the posteromedial border of the distal two thirds of the tibia and is associated with activity” (Lines 47-49).
Ln: 46: The statement may be incorrect as the ref 45 used in this paper: is on the risk factors of MTSS in runners- Systematic review
We are referring to systematic reviews specific of novice and recreational runners, and that review we cite is focused on professional runners.
Materials:
Good description flow
Studies included seem not to cover the whole spectrum: eg Ref 45 is not included. Similarly other studies from literature are missed
We have repeated and updated the search procedure. We have also added a search in the database CINAHL. Review papers have been considered to be exclusion criteria.
Inclusion criteria in this study had:
"All participants in all papers were runners or played sports that were running-based."
Results
Ln 155: Etiological factors:
Extrinsic factors not described which form an important component of risk factors in MTSS
We have described only the factors mentioned in the studies included, and they are mainly focused on biomechanical factors.
3.3 Diagnostic procedures:
no description on History and Physical examination
You are right. We have changed the sentence referred to history and physical examination, and we have added it to the discussion (line 285). We do not intend to describe these diagnostic protocols extensively, but we only mention these methods to indicate that they are important instruments to diagnose medial tibial stress syndrome.
comment on CT are misleading as the author have not commented on any diagnostic signs of MTSS on CT.
We have added a sentence in order to explain it: “Although more studies are required, CT may be useful to detect stress-induced bone remodelling” (line 295). The diagnostic sign observed through computed tomography in subjects with medial tibial stress syndrome is the stress-induced bone remodelling, which could precede a stress fracture.
Osteopenia in runners? what's the prevalence in this population
We have eliminated this sentence about osteopenia: “Good diagnostic accuracy was demonstrated for high-resolution CT and, although more studies are required, CT may be useful to detect stress-induced bone remodelling” (line 294)
Pressure altimeter: is it diagnostic?
Thank you for your question. Pressure algometry is not a diagnostic method of medial tibial stress syndrome. We have included among the diagnostic procedures because this technique is useful to assess pain in subjects with this injury. Thus, pressure algometry may be a proper tool to monitor the rehabilitation process and the injury state.
Ln214: confusing line, CT examination on eleven runners, twenty having no symptoms"
We have modified the sentence to clarify it: “Gaeta et al [25] carried out CT examinations on eleven runners with MTSS, twenty asymptomatic runners, and control peers not involved in a sport,…” (line 293).
Ln 217: to make a conclusion after one study
This is true. We have changed the sentence to smooth the findings: “although more studies are required, CT may be useful to detect stress-induced bone remodelling”(line 294).
Ln219- 222: comparison is being made with athletes to derive conclusion in non athletes---inadequate
In this particular case (diagnostic procedures) we believe that comparisons between athletes and non-athletes may be useful. Thus, since the injury itself is the same either in professional or in non-professional runners (although risk factors may differ, the physiopathology of the injury does not vary), we consider that it can be interesting to mention. In fact, the study by Magnusson et al. (Magnusson, H.I.; Westlin, N.E.; Nyqvist, F.; Gärdsell, P.; Seeman, E.; Karlsson, M.K. Abnormally decreased regional bone density in athletes with medial tibial stress syndrome. Am. J. Sports Med. 2001, 29, 712–715.) includes different sport practitioners, as soccer, handball, badminton and basketball players, weightlifters, and long-distance runners, who train a mean of 7h per week.
Treatment: inadequate and Incomplete methods: is this because inappropriate study inclusion?
In relation to our sentence “However, there is currently a lack of evidence to allow decisions about which techniques are the most appropriate for rehabilitation and re-adaptation [57].” (line 310), we have based on the systematic review by Winters et al. (Winters, M.; Bakker, E.W.P.; Moen, M.H.; Barten, C.C.; Teeuwen, R.; Weir, A. Medial tibial stress syndrome can be diagnosed reliably using history and physical examination. Br. J. Sports Med. 2018, 52, 1267–1272.). However, the RCT included have a level of evidence III, and the non-RCT have a level of evidence of IV. These poor levels of evidence might prevent conclude solid evidences.
orthotics: Can we generalise the advice?- unlikely
What about extrinsic factor management
what about systemic reasons
Foot orthoses are able to reduce foot eversion. Therefore, they may be a good tool to correct over-pronation. We have modified the sentence: “so this method may be an effective alternative to treat MTSS in recreational runners [30]. Nevertheless, the normalized foot-pressure distribution patterns during running through arch-support orthoses do not involve a reduction in the level of pain in these subjects”. (line 321).
We are aware that it is possible that factors which are not taking into account may contribute to the ineffectiveness of this method.
The authors mention about osteopenia-- what about its management
As we have mentioned above, we have deleted the sentence about osteopenia, in order to avoid any misleading conclusion.
Conclusion:
Misleading especially with diagnosis with CT
The author talks bout Hx an physical management-- but no specific mention in the text
SWT: no study quoted for convincing evidence on this treatment
What
We have rewritten the conclusion: “The main factors for the development of MTSS in novice and recreational runners seem to be intrinsic and are commonly related to a biomechanical origin. To achieve a diagnosis, CT scan has shown to be an accurate method, detecting the stress-induce bone remodelling. Pressure algometry is a complement tool for assessing pain, injury state and rehabilitation process. The approximate recovery time is sixteen to eighteen weeks. Treatment methods include arch-support orthoses, which induce positive effects on the foot-pressure distribution, and SWT, which might be able to reduce the level of pain. Nevertheless, we have to bearing in mind the poor methodological quality of the included studies, which prevents us to obtain solid evidence about MTSS in novice and recreational runners”.
Reviewer 2 Report
MTSS in novice and recreational runners: A systematic review
A systematic review is a review of a clearly formulated question that uses systematic and explicit methods to identify, select, and critically appraise relevant research, and to collect and analyze data from the studies that are included in the review (Moher et al., 2009).
The question in this review is not clearly formulated. It reports the intention of the review is to “assess and summarise the current literature about MTSS in novice and recreational runners”. There is a plethora of recent studies on this exact topic, some of which are referenced within this review. The reader needs to see exactly what this review is adding to the current knowledge. The definition of novice and recreational runners has not been clearly defined. The introduction, line 45, states “recreational runners are defined as someone with over three months running experience”. Really?? I would have thought distance ran per week would be more appropriate. Within the method, line 65, it reports “those not competing at a national or international level”. Line 39 of the introduction states MTSS is caused by over-use. Why therefore have the authors been able to define this targeted group of this review using three months running experience and not national or international athletes? These definitions are not associated caused of MTSS – once again I suggest distance ran per week as more appropriate and this is not clearly identified in each of the included studies.
The question of this review needs to be clearly formulated, with clear definitions for novice and recreational runners, so the reader can clearly discern these outcomes differ from the current knowledge.
As with all research, the value of a systematic review depends on what was done, what was found, and the clarity of reporting. As with other publications, the reporting quality of systematic reviews varies, limiting readers’ ability to assess the strengths and weaknesses of those studies (Moher et al., 2009).
This review has used two researchers to assess the quality of the included studies. PEDro scale for the one RCT included and the Newcastle-Ottawa Scale for the 4 case control studies, 3 cohort studies and 1 cross sectional studies included. Two descriptive studies were also included. It is important to indicate to the reader the Levels of Evidence for each of the included studies using the NHMRC or Oxford Centre for Evidence Based Medicine (CEBM). Most of the include studies are not a high level of evidence. This must be highlighted within the discussion, with reasons given.
Apart from graphically including the results of the critical appraisal of each study there is no mention of these results within the discussion. The outcomes for each study are written up as if they are absolutely accurate and can be relied upon with no discussion regarding the types of bias or failures with internal or external validity. Some studies scored poorly and hence concluding their outcomes as valuable information is incorrect.
Discussion of the outcomes of the critical appraisal and Levels of Evidence must be included in the discussion along with a limitations section clearly detailing the shortfalls of the findings of this review.
Moher, D., Liberati, A., Tetzlaff, J., Altman, D., & The PRISMA Group. (2009, July 21). Preferred reporting items for systematic reviews and meta-analyses: The PRISMA statement. PLoS Medicine, 6(7): e1000097. https://doi.org/10.1371/journal.pmed.1000097
Author Response
MTSS in novice and recreational runners: A systematic review
A systematic review is a review of a clearly formulated question that uses systematic and explicit methods to identify, select, and critically appraise relevant research, and to collect and analyze data from the studies that are included in the review (Moher et al., 2009).
The question in this review is not clearly formulated. It reports the intention of the review is to “assess and summarise the current literature about MTSS in novice and recreational runners”. There is a plethora of recent studies on this exact topic, some of which are referenced within this review. The reader needs to see exactly what this review is adding to the current knowledge.
Thank you for your analysis. We pretend to cover the whole spectrum of existing studies on medial tibial stress syndrome in novice and recreational runners, without focus on a particular topic (i.e. epidemiology). As far as we known, the current literature does not count with a systematic review about MTSS in this population, since as we mention in the introduction, the studies on novice and recreational runners are scarce and the vast majority only include professional runners/athletes.
Therefore, since no reviews have been performed on this exact topic, we think that it is interesting given the large population of novice and recreational runners year by year. The running population is on the rise, especially in long distance events, with the risk of increase of running-related injuries. Novice runners seem to be the most vulnerable group, with a much higher risk of injury (Videbaek S, Bueno AM, Nielsen RO, Rasmussen S. Incidence of running-related injuries per 1000 h of running in different types of runners: a systematic review and meta-analysis. Sports Med 2015; 45:1017-1026).
We are aware that treatment methods may be similar for professional and non-professional runners, but as our initial purpose was to cover the entire studies specific of novice and recreational runners, we have considered maintaining these studies.
The definition of novice and recreational runners has not been clearly defined. The introduction, line 45, states “recreational runners are defined as someone with over three months running experience”. Really?? I would have thought distance ran per week would be more appropriate. Within the method, line 65, it reports “those not competing at a national or international level”. Line 39 of the introduction states MTSS is caused by over-use. Why therefore have the authors been able to define this targeted group of this review using three months running experience and not national or international athletes? These definitions are not associated caused of MTSS – once again I suggest distance ran per week as more appropriate and this is not clearly identified in each of the included studies. The question of this review needs to be clearly formulated, with clear definitions for novice and recreational runners, so the reader can clearly discern these outcomes differ from the current knowledge.
You are right. First, we apologize for including the sentence “those not competing at a national or international level”; it was a mistake and we have deleted it.
Regarding the definition of recreational runners, the aforementioned well-designed systematic review and meta-analysis by Videbæk et al. establishes a cut-off around 8-13 weeks of running experience for distinguishing a novice runner from a recreational runner. We have based on this study and we have establish a 3-month running experience as a criteria for be considered a novice or recreational runner.
We do not know if distance ran per week can be used for this purpose, since we have no fond literature about it.
As with all research, the value of a systematic review depends on what was done, what was found, and the clarity of reporting. As with other publications, the reporting quality of systematic reviews varies, limiting readers’ ability to assess the strengths and weaknesses of those studies (Moher et al., 2009).
This review has used two researchers to assess the quality of the included studies. PEDro scale for the one RCT included and the Newcastle-Ottawa Scale for the 4 case control studies, 3 cohort studies and 1 cross sectional studies included. Two descriptive studies were also included. It is important to indicate to the reader the Levels of Evidence for each of the included studies using the NHMRC or Oxford Centre for Evidence Based Medicine (CEBM). Most of the include studies are not a high level of evidence. This must be highlighted within the discussion, with reasons given.
We have reviewed the included articles and we have redone the quality appraisal. Furthermore, we have included the NHMRC levels of evidence, and we have added it to the discussion.
Apart from graphically including the results of the critical appraisal of each study there is no mention of these results within the discussion. The outcomes for each study are written up as if they are absolutely accurate and can be relied upon with no discussion regarding the types of bias or failures with internal or external validity. Some studies scored poorly and hence concluding their outcomes as valuable information is incorrect.
Discussion of the outcomes of the critical appraisal and Levels of Evidence must be included in the discussion along with a limitations section clearly detailing the shortfalls of the findings of this review.
Thank you very much. We have included a section explaining the lower evidence of the great majority of the studies and the existing limitations, which prevent us to obtain solid evidence.
“The studies included in this systematic review present some design biases which prevent us to establish solid conclusions. It is worth mentioning that only the study by Newman et al, 2017 [31] is randomized controlled trial (level of evidence II), while the rest of the studies have a lower level of evidence. Therefore, when interpreting the results, we have to be aware of this circumstance, which is a major limitation in order to obtain solid evidence from this systematic review. We encourage further high-quality research on this specific population to provide clear conclusions” (line 342).
Reviewer 3 Report
Review for International Journal of Environmental Research and Public Health
Thank you for the opportunity to review this article. The language and grammar of the article is quite good and it was easy to read. I think the topic is of interest and important to clinicians such as medical specialists, physiotherapists and podiatrists. However, I feel that there is a substantial component of the quality assessment missing and that the combination of results and discussion sections may mislead readers. In relation to this the discussion seems heavily biased to the finding of MTSS aetiological factors and management being related to foot over-pronation which I do not feel is substantiated by the results.
With relation to the quality assessment, it is good to see that quality assessment using recognised tools was undertaken. However, apart from presentation of the application of the tools as a table, there was no discussion around the overall quality of the articles and the short-comings of the methods used. This is particularly important if it needs to be discussed in relation to the findings of the research (i.e. if two different findings are found relating to treatment, but one article is of higher quality, then the findings are subsequently more dependable). There was also no quality assessment done on two of the articles, as they seemingly didn’t suit the tool used. I do not see any reason why they could not have a quality assessment undertaken, even if another tool needed to be utilised. The Loudon & Dolphino study (25) for example is quite an important observational study (utilising an intervention/s) and should be assessed for quality if we are to consider the treatment findings appropriately.
Specific feedback can be found below:
Introduction:
Line 39 – The definition of MTSS needs a reference.
Some more information about who it affects would be beneficial, i.e. you mention that the incidence is higher in recreational athletes, but it would also be good to point out which age category or sex predilection exists for the condition.
Methods:
I am concerned, considering the focus on novel and recreational runners, that the use of the term ‘athletes’ in the search terms will exclude some studies for this population. This seems to be a mismatch, even in the introduction you introduce the term ‘athlete’s’ when referring to professionals, but ‘runners’ when discussing novice and recreational.
I also feel that the search could be somewhat more extensive, including the databases AMED, Cinahl and Google Scholar, as well as utilising methods such as ‘pearling’ the reference lists of included studies and previous systematic reviews.
As outlined above, I feel that there needs to be a quality appraisal undertaken on all of the studies included and do not see why this cannot be the case.
There was no information as to how any discrepancies or ambiguity regarding the quality assessment was managed between assessors. It is also common practice to outline the initials of the authors responsible for each of the phases including the quality assessment.
Results and Discussion:
As outlined above, I think it is important for the results and discussions to be separated. Predominantly so that readers can better determine the actual findings with the interpretation made by the authors. It also detracts from the findings by introducing other references, not specific to the review itself and the quality assessment of the articles.
In figure 1, you state that 729 articles were excluded after title/abstract assessment, but in-text state the number as 731. Why the discrepancy?
Line 162 – 164 – this needs to go to a separate discussion section as it is very much an interpretation of the results. I also feel this is overrepresenting the findings. If anything I would say that increased pelvic drop seems to be the more crucial predisposing factor. You are suggesting that over-pronation is the factor even though almost none of them measured or assessed this. And some of the associations you mentioned relating to over-pronation are the cause of increased pronation rather than the consequence..i.e early heel lift. Most of the findings you are associating with over-pronation may be related, but actually lack strong relationships, there are other causes… i.e. whilst pronation may also be related to internal rotation of the leg, there are also many many other causes of this (such as hip related things). So unless they are directly measuring this, I feel you need to be careful in how you are stating over-pronation as an aetiological cause. In relation to this, one of the studies where they do consider pronation, by measuring FPI, firstly this is a static measure not a dynamic measure so you have to be careful again about over-stating the role of pronation as the posture may differ to the kinematics, and secondly the only aspect of significance was the difference between neutral and resting calcaneal position.
Line 201 – I think this is an interesting finding considering you have not discussed any of the literature which has looked at training technique, in particular gait retraining such as: Sharma, J., Weston, M., Batterham, A., & Spears, I. (2014). Gait retraining and incidence of medial tibial stress syndrome in army recruits. Medicine & Science in Sports & Exercise, 46(9), 1684-1692.
I realise your review doesn’t include army recruits, but you shouldn’t ignore that this has been considered if it doesn’t match your target population, as in this instance I believe the findings would still apply.
Line 204 – 206 – this is more suited to the introduction rather than the findings
Line 217 – 219 – this should also be in the discussion separately
Line 237 – I do not understand what is meant by ‘improve footprint’.
Line 237 – what you mean by “impulse” is not clear
Line 241 – Should this read “normalised foot-pressure distribution patterns”
Line 242 – “improvement in pain” is ambiguous, do you mean reduction in the level of pain? Improvement could mean change in the duration of pain, level of pain or frequency of pain.
Line 248 – It would be good to clarify what type of shockwave therapy was applied (i.e. radial?)
Line 250 – 251 – the findings relating to plantar fasciitis is not relevant here, particularly given they are such different pathologies.
Line 255 – 258 – this is an over-generalisation of the results
Conclusion:
I find this is non-specific and doesn’t relate well to the specific findings of the systematic review. At least nothing you have discussed in your ‘discussion’. I.e. you haven’t even mentioned the balance of intrinsic versus extrinsic factors until now. You have not done any analysis on this or compared the quality, significance nor applicability of the findings related to the papers. I.e. were all the papers investigating intrinsic factors of poor quality? Or did they have smaller numbers? Do you think that it is purely because the research articles weren’t investigating extrinsic factors…i.e. if they aren’t evaluated then it doesn’t mean they are not there.
Tables and Figures:
Table 2 – needs to be reformatted to see the full-text of the study design
It would be good to include the year of publication in the table (maybe after each reference)
It would be useful to have labels again with the numbers, i.e. 1. Case control – Selection…… 2. Cohort – Selection….
Maybe for the Cross -sectional studies, outcome section state that 3) is N/A
Table 3 – 5: It would be good reformat and put in landscape to improve readability.
I also think the year’s of the articles are worth including
A separate column for the participant numbers would be good (just to tease out this detail as it is very important to the generalisability of the findings)
I also think that a separate column for the outcomes/measures would be good to make it clearer rather than the large amount of text in the methodology column
It would also be good to have more detail around the level of significance i.e. can the actual p-values be included?
I think a lot of the text can be reduced from the tables I.e “Participants were considered to have successful intervention based on a 50% improvement on the final NPRS”, could read “Successful intervention on 50% improvement on final NPRS”
Table 3 – Not sure what ‘Craig test’ is, can this please be defined?
In the tables you swap between using > greater than, and á increased. I think that in some cases it would be better for you to use á i.e. MTSS has >prevalence rate…might read better as MTSS has á(increased) prevalence rate.
I don’t think you need to define > < á or VS as they are internationally recognised symbols and abbreviations.
Table 5 – Need to include ND definition
Appendices:
It is often common practice to list the excluded articles and their reasons specifically in an appendix. This is important for people to check when they feel an article should have been included if it was not. It also then easier to determine if the article was missed through the initial search or was specifically excluded from the analysis.
Author Response
Thank you for the opportunity to review this article. The language and grammar of the article is quite good and it was easy to read. I think the topic is of interest and important to clinicians such as medical specialists, physiotherapists and podiatrists. However, I feel that there is a substantial component of the quality assessment missing and that the combination of results and discussion sections may mislead readers. In relation to this the discussion seems heavily biased to the finding of MTSS aetiological factors and management being related to foot over-pronation which I do not feel is substantiated by the results.
With relation to the quality assessment, it is good to see that quality assessment using recognised tools was undertaken. However, apart from presentation of the application of the tools as a table, there was no discussion around the overall quality of the articles and the short-comings of the methods used. This is particularly important if it needs to be discussed in relation to the findings of the research (i.e. if two different findings are found relating to treatment, but one article is of higher quality, then the findings are subsequently more dependable). There was also no quality assessment done on two of the articles, as they seemingly didn’t suit the tool used. I do not see any reason why they could not have a quality assessment undertaken, even if another tool needed to be utilised. The Loudon & Dolphino study (25) for example is quite an important observational study (utilising an intervention/s) and should be assessed for quality if we are to consider the treatment findings appropriately.
Specific feedback can be found below:
Introduction:
Line 39 – The definition of MTSS needs a reference.
You are right. We have included a reference for the definition of medial tibial stress syndrome (Mubarak SJ, Gould RN, Lee YF, Schmidt DA, Hargens AR. The medial tibial stress syndrome. A cause of shin splints. Am J Sports Med. 1982;10(4):201-205. doi:10.1177/036354658201000402).
Some more information about who it affects would be beneficial, i.e. you mention that the incidence is higher in recreational athletes, but it would also be good to point out which age category or sex predilection exists for the condition.
We agree with your suggestion. Unfortunately, the current literature does not describe differences in incidence and risk factors between men and women in depth, and the age ranges are to wide (18-65 years) and does not allow solid conclusions to be drawn (Buist I, Bredeweg SW, Lemmink KA, van Mechelen W, Diercks RL. Predictors of running-related injuries in novice runners enrolled in a systematic training program: a prospective cohort study. Am J Sports Med. 2010;38(2):273-280. doi:10.1177/0363546509347985).
Methods:
I am concerned, considering the focus on novel and recreational runners, that the use of the term ‘athletes’ in the search terms will exclude some studies for this population. This seems to be a mismatch, even in the introduction you introduce the term ‘athlete’s’ when referring to professionals, but ‘runners’ when discussing novice and recreational.
I also feel that the search could be somewhat more extensive, including the databases AMED, Cinahl and Google Scholar, as well as utilising methods such as ‘pearling’ the reference lists of included studies and previous systematic reviews.
Thank you for your advice. We have updated the search (July 2020) and we have added the terms “novice” and “recreational”. Furthermore, we have also performed the search in CINAHL, as you have suggested, and with manually screened references of the included articles and citation tracking of included studies in Scopus.
As outlined above, I feel that there needs to be a quality appraisal undertaken on all of the studies included and do not see why this cannot be the case.
We have performed the quality appraisal for all the studies included. We have considered the study by Loudon and Dolphino as an interventional study (case series), and the study by Raissi et al. as a cohort study, although it is not a properly said cohort study, since there are no exposed and non-exposed individuals, and there is a unique sample. We have used the PEDro and the Newcastle-Ottawa scales to assess them, respectively.
There was no information as to how any discrepancies or ambiguity regarding the quality assessment was managed between assessors. It is also common practice to outline the initials of the authors responsible for each of the phases including the quality assessment.
We had included this statement: “Any disagreements between reviewers were settled through discussion with a third reviewer” (line 72). We have now specified the authors (initials) enrolled in each of the study phases.
Results and Discussion:
As outlined above, I think it is important for the results and discussions to be separated. Predominantly so that readers can better determine the actual findings with the interpretation made by the authors. It also detracts from the findings by introducing other references, not specific to the review itself and the quality assessment of the articles.
We are aware that this point depends on individual preferences. We have selected this journal specifically because it allows combining the results and the discussion sections, as it points out in the author guidelines. Based on this and bearing in mind the structure of our review, which is heterogeneous regarding the thematic (diagnostic procedures, treatment methods, etc), we firmly believe that combining both sections is the optimal way to facilitate reading and understanding.
In figure 1, you state that 729 articles were excluded after title/abstract assessment, but in-text state the number as 731. Why the discrepancy?
We apologize for the mistake. We have amended it and we have updated the search.
Line 162 – 164 – this needs to go to a separate discussion section as it is very much an interpretation of the results. I also feel this is overrepresenting the findings. If anything I would say that increased pelvic drop seems to be the more crucial predisposing factor. You are suggesting that over-pronation is the factor even though almost none of them measured or assessed this. And some of the associations you mentioned relating to over-pronation are the cause of increased pronation rather than the consequence..i.e early heel lift. Most of the findings you are associating with over-pronation may be related, but actually lack strong relationships, there are other causes… i.e. whilst pronation may also be related to internal rotation of the leg, there are also many many other causes of this (such as hip related things). So unless they are directly measuring this, I feel you need to be careful in how you are stating over-pronation as an aetiological cause. In relation to this, one of the studies where they do consider pronation, by measuring FPI, firstly this is a static measure not a dynamic measure so you have to be careful again about over-stating the role of pronation as the posture may differ to the kinematics, and secondly the only aspect of significance was the difference between neutral and resting calcaneal position.
We agree with your point of view, and we have tried to soften the role of over-pronation in the development of medial tibial stress syndrome in relation to the other biomechanical predisposing factors. In this regard, we have specified that over-pronation appears to be related with the development of this pathology, since all the biomechanical factors that are frequently involved (e-g. navicular drop, early heel lift, etc.) are commonly associated with over-pronation of the foot. Nevertheless, we do not mean to say that pronation is the most relevant predisposing factor and it is the cause of the rest of biomechanical dysfunctions, but we describe this circumstance as an important factor associated with a plethora of biomechanical elements. Therefore, despite we give prominence to over-pronation, we do not state that it is neither cause nor consequence of medial tibial stress syndrome.
Line 201 – I think this is an interesting finding considering you have not discussed any of the literature which has looked at training technique, in particular gait retraining such as: Sharma, J., Weston, M., Batterham, A., & Spears, I. (2014). Gait retraining and incidence of medial tibial stress syndrome in army recruits. Medicine & Science in Sports & Exercise, 46(9), 1684-1692.
I realise your review doesn’t include army recruits, but you shouldn’t ignore that this has been considered if it doesn’t match your target population, as in this instance I believe the findings would still apply.
That’s right. We have included this study in the discussion, as follows:
“Indeed, army recruits at risk of MTSS who were submitted to a 26 week-gait retraining protocol consisting on plantar pressure system biofeedback and verbal orders and corrections had lower incidence rate of MTSS at the end of the protocol. It is also worth noting that this procedure was complemented with strength, flexibility and neuromuscular control exercises, evidencing the importance of designing multicomponent protocols for this purpose” (line 276).
Line 204 – 206 – this is more suited to the introduction rather than the findings
We have included it, in order to introduce and enrich the discussion.
Line 217 – 219 – this should also be in the discussion separately
As we mentioned above, we have elaborated the manuscript according with the journal guidelines.
Line 237 – I do not understand what is meant by ‘improve footprint’.
We agree with that. We have changed the term and we have substituted it by “foot pressure” (line 318).
Line 237 – what you mean by “impulse” is not clear
“A mechanical impulse is the momentum imparted by the product of force and the time duration of its action”. (C Nevin. University of Cape Town, 1995. 1, 1995. Initiation and control of gait from first principles: a mathematically animated model of the foot).
Line 241 – Should this read “normalised foot-pressure distribution patterns”
Thank you for the correction. We have changed it.
Line 242 – “improvement in pain” is ambiguous, do you mean reduction in the level of pain? Improvement could mean change in the duration of pain, level of pain or frequency of pain.
We agree with your appreciation. We have modified it.
Line 248 – It would be good to clarify what type of shockwave therapy was applied (i.e. radial?)
We have specified that the study refers to focused shockwave therapy.
Line 250 – 251 – the findings relating to plantar fasciitis is not relevant here, particularly given they are such different pathologies.
According to your suggestion, we have deleted this sentence.
Line 255 – 258 – this is an over-generalisation of the results
Thank you very much for your advice. Our message was ambiguous, but we have tried to explain it in more detail: “Patients with MTSS have seen faster recovery and lesser level of pain following SWT compared to exercise protocols [63,64]…” (line 332).
Conclusion:
I find this is non-specific and doesn’t relate well to the specific findings of the systematic review. At least nothing you have discussed in your ‘discussion’. I.e. you haven’t even mentioned the balance of intrinsic versus extrinsic factors until now. You have not done any analysis on this or compared the quality, significance nor applicability of the findings related to the papers. I.e. were all the papers investigating intrinsic factors of poor quality? Or did they have smaller numbers? Do you think that it is purely because the research articles weren’t investigating extrinsic factors…i.e. if they aren’t evaluated then it doesn’t mean they are not there.
We have rewritten the conclusion based on the data collected by the included articles, and we have tried to avoid any speculation or ambiguity. We have also added a paragraph in relation to the methodological quality of the studies: “Nevertheless, we have to bearing in mind the poor methodological quality of the included studies, which prevents us to obtain solid evidence about MTSS in novice and recreational runners.” (line 356).
Tables and Figures:
Table 2 – needs to be reformatted to see the full-text of the study design
It would be good to include the year of publication in the table (maybe after each reference)
It would be useful to have labels again with the numbers, i.e. 1. Case control – Selection…… 2. Cohort – Selection….
Maybe for the Cross -sectional studies, outcome section state that 3) is N/A
Thank you for your advices. We have reformatted the table according to your instructions.
Table 3 – 5: It would be good reformat and put in landscape to improve readability.
I also think the year’s of the articles are worth including
A separate column for the participant numbers would be good (just to tease out this detail as it is very important to the generalisability of the findings)
I also think that a separate column for the outcomes/measures would be good to make it clearer rather than the large amount of text in the methodology column
We have included the year of the studies. Bearing in mind that the format of the submission requires vertical pages, it is no possible to include so many columns, and we have to focalize the information in a short space.
It would also be good to have more detail around the level of significance i.e. can the actual p-values be included?
Thank you for your suggestion. In the beginning we tried to do it like that, but after a few attempts we have considered that it is more appropriate to group all the statistically significant results (p<0.05) in the same column, since place the individual p value could be more confusing for the reader, due to the little space we have and the amount of text. In this way, the reader interpret from the beginning that the results are all statistically significant. If the results are no significant, we have specified that there were no statistically significant differences.
I think a lot of the text can be reduced from the tables I.e “Participants were considered to have successful intervention based on a 50% improvement on the final NPRS”, could read “Successful intervention on 50% improvement on final NPRS”
We have changed this sentence based on your directions (table 5)
Table 3 – Not sure what ‘Craig test’ is, can this please be defined?
Craig’s test is the most commonly used physical method for measuring femoral anteversion (Choi BR, Kang SY. Intra- and inter-examiner reliability of goniometer and inclinometer use in Craig's test. J Phys Ther Sci. 2015;27(4):1141-1144.).
Nevertheless we have deleted it from the table and we have substituted it by “femoral anteversion”, to facilitate the compression without need to consult external sources.
In the tables you swap between using > greater than, and á increased. I think that in some cases it would be better for you to use á i.e. MTSS has >prevalence rate…might read better as MTSS has á(increased) prevalence rate.
I don’t think you need to define > < á or VS as they are internationally recognised symbols and abbreviations.
We have deleted >/< symbols and we have replaced them by ↑/↓ .
Table 5 – Need to include ND definition
Done.
Appendices:
It is often common practice to list the excluded articles and their reasons specifically in an appendix. This is important for people to check when they feel an article should have been included if it was not. It also then easier to determine if the article was missed through the initial search or was specifically excluded from the analysis.
Reviewer is right, that in some occasions and depending on the journal this information is available. However, we have missed this point, and we have not recorded this information, since we consider in the beginning that providing the term of searching (Appendix 1) would be enough and satisfy the scientific accuracy.
Round 2
Reviewer 1 Report
as advised before
Reviewer 2 Report
Thank you for taking the time to edit your manuscript.
I am still not convinced it adds any new information to our current knowledge but I will leave that to the reader to deduce.